# PixelDefend: Leveraging Generative Models to Understand and Defend against Adversarial Examples

**Yang Song**
Stanford University
yangsong@cs.stanford.edu

**Taesup Kim**
Université de Montréal
taesup.kim@umontreal.ca

**Sebastian Nowozin**
Microsoft Research
nowozin@microsoft.com

**Stefano Ermon**
Stanford University
ermon@cs.stanford.edu

**Nate Kushman**
Microsoft Research
nkushman@microsoft.com

## Abstract

Adversarial perturbations of normal images are usually imperceptible to humans, but they can seriously confuse state-of-the-art machine learning models. What makes them so special in the eyes of image classifiers? In this paper, we show empirically that adversarial examples mainly lie in the low probability regions of the training distribution, regardless of attack types and targeted models. Using statistical hypothesis testing, we find that modern neural density models are surprisingly good at detecting imperceptible image perturbations. Based on this discovery, we devised *PixelDefend*, a new approach that *purifies* a maliciously perturbed image by moving it back towards the distribution seen in the training data. The purified image is then run through an unmodified classifier, making our method agnostic to both the classifier and the attacking method. As a result, PixelDefend can be used to protect already deployed models and be combined with other model-specific defenses. Experiments show that our method greatly improves resilience across a wide variety of state-of-the-art attacking methods, increasing accuracy on the strongest attack from 63% to 84% for Fashion MNIST and from 32% to 70% for CIFAR-10.

## 1 Introduction

Recent work has shown that small, carefully chosen modifications to the inputs of a neural network classifier can cause the model to give incorrect labels (Szegedy et al., 2013; Goodfellow et al., 2014). This weakness of neural network models is particularly surprising because the modifications required are often imperceptible, or barely perceptible, to humans. As deep neural networks are being deployed in safety-critical applications such as self-driving cars (Amodei et al., 2016), it becomes increasingly important to develop techniques to handle these kinds of inputs.

**Rethinking adversarial examples** The existence of such adversarial examples seems quite surprising. A neural network classifier can get super-human performance (He et al., 2015) on clean test images, but will give embarrassingly wrong predictions on the same set of images if some imperceptible noise is added. What makes this noise so special to deep neural networks?

In this paper, we propose and empirically evaluate the following hypothesis: Even though they have very small deviations from clean images, adversarial examples largely lie in the low probability regions of the distribution that generated the data used to train the model. Therefore, they fool

classifiers mainly due to covariate shift. This is analogous to training models on MNIST (LeCun et al., 1998) but testing them on Street View House Numbers (Netzer et al., 2011).

To study this hypothesis, we first need to estimate the probability density of the underlying training distribution. To this end, we leverage recent developments in generative models. Specifically, we choose a PixelCNN (van den Oord et al., 2016b) model for its state-of-the-art performance in modeling image distributions (van den Oord et al., 2016a; Salimans et al., 2017) and tractability of evaluating the data likelihood. In the first part of the paper, we show that a well-trained Pixel-CNN generative model is very sensitive to adversarial inputs, typically giving them several orders of magnitude lower likelihoods compared to those of training and test images.

**Detecting adversarial examples**  An important step towards handling adversarial images is the ability to detect them. In order to catch any kind of threat, existing work has utilized confidence estimates from Bayesian neural networks (BNNs) or dropout (Li & Gal, 2017; Feinman et al., 2017). However, if their model is misspecified, the uncertainty estimates can be affected by covariate shift (Shimodaira, 2000). This is problematic in an adversarial setting, since the attacker might be able to make use of the inductive bias from the misspecified classifier to bypass the detection.

Protection against the strongest adversary requires a pessimistic perspective—our assumption is that the classifier cannot give reliable predictions for any input outside of the training distribution. Therefore, instead of relying on label uncertainties given by the classifier, we leverage statistical hypothesis testing to detect any input not drawn from the same distribution as training images.

Specifically, we first compute the probabilities of all training images under the generative model. Afterwards, for a novel input we compute the probability density at the input and evaluate its rank (in ascending order) among the density values of all training examples. Next, the rank can be used as a test statistic and gives us a $p$-value for whether or not the image was drawn from the training distribution. This method is general and practical and we show that the $p$-value enables us to detect adversarial images across a large number of different attacking methods with high probability, even when they differ from clean images by only a few pixel values.

**Purifying adversarial examples**  Since adversarial examples are generated from clean images by adding imperceptible perturbations, it is possible to decontaminate them by searching for more probable images within a small distance of the original ones. By limiting the $L^\infty$ distance[1], this image *purification* procedure generates only imperceptible modifications to the original input, so that the true labels of the purified images remain the same. The resulting purified images have higher probability under the training distribution, so we can expect that a classifier trained on the clean images will have more reliable predictions on the purified images. Moreover, for inputs which are not corrupted by adversarial perturbations the purified results remain in a high density region.

We use this intuition to build *PixelDefend*, an image purification procedure which requires no knowledge of the attack nor the targeted classifier. PixelDefend approximates the training distribution using a PixelCNN model. The constrained optimization problem of finding the highest probability image within an $\epsilon$-ball of the original is computationally intractable, however, so we approximate it using a greedy decoding procedure. Since PixelDefend does not change the classification model, it can be combined with other adversarial defense techniques, including adversarial training (Goodfellow et al., 2014), to provide synergistic improvements. We show experimentally that PixelDefend performs exceptionally well in practice, leading to state-of-the art results against a large number of attacks, especially when combined with adversarial training.

**Contributions**  Our main contributions are as follows:

- We show that generative models can be used for detecting adversarially perturbed images and observe that most adversarial examples lie in low probability regions.
- We introduce a novel family of methods for defending against adversarial attacks based on the idea of purification.
- We show that a defensive technique from this family, PixelDefend, can achieve state-of-the-art results on a large number of attacking techniques, improving the accuracy against the strongest adversary on the CIFAR-10 dataset from 32% to 70%.

---

[1] We note that there are many other ways of defining distance of images. In this paper we use $L^\infty$ norm.

## 2 BACKGROUND

### 2.1 ATTACKING METHODS

Given a test image $\mathbf{X}$, an attacking method tries to find a small perturbation $\mathbf{\Delta}$ with $\|\mathbf{\Delta}\|_\infty \leq \epsilon_{\text{attack}}$, such that a classifier $f$ gives different predictions on $\mathbf{X}^{adv} \triangleq \mathbf{X} + \mathbf{\Delta}$ and $\mathbf{X}$. Here colors in the image are represented by integers from 0 to 255. Each attack method is controlled by a configurable $\epsilon_{\text{attack}}$ parameter which sets the maximum perturbation allowed for each pixel in integer increments on the color scale. We only consider white-box attacks in this paper, *i.e.*, the attack methods can get access to weights of the classifier. In the following, we give an introduction to all the attacking methods used in our experiments.

**Random perturbation (RAND)** Random perturbation is arguably the weakest attacking method, and we include it as the simplest baseline. Formally, the randomly perturbed image is given by

$$\mathbf{X}^{adv} = \mathbf{X} + \mathcal{U}(-\lfloor\epsilon_{\text{attack}}\rfloor, \lfloor\epsilon_{\text{attack}}\rfloor),$$

where $\mathcal{U}(a, b)$ denotes an element-wise uniform distribution of integers from $[a, b]$.

**Fast gradient sign method (FGSM)** Goodfellow et al. (2014) proposed the generation of malicious perturbations in the direction of the loss gradient $\nabla_\mathbf{X} L(\mathbf{X}, y)$, where $L(\mathbf{X}, y)$ is the loss function used to train the model. The adversarial examples are computed by

$$\mathbf{X}^{adv} = \mathbf{X} + \epsilon_{\text{attack}} \operatorname{sign}(\nabla_\mathbf{X} L(\mathbf{X}, y)).$$

**Basic iterative method (BIM)** Kurakin et al. (2016) tested a simple variant of the fast gradient sign method by applying it multiple times with a smaller step size. Formally, the adversarial examples are computed as

$$\mathbf{X}_0^{adv} = \mathbf{X}, \quad \mathbf{X}_{n+1}^{adv} = \operatorname{Clip}_\mathbf{X}^{\epsilon_{\text{attack}}} \left\{ \mathbf{X}_n^{adv} + \alpha \operatorname{sign}(\nabla_\mathbf{X} L(\mathbf{X}_n^{adv}, y)) \right\},$$

where $\operatorname{Clip}_\mathbf{X}^{\epsilon_{\text{attack}}}$ means we clip the resulting image to be within the $\epsilon_{\text{attack}}$-ball of $\mathbf{X}$. Following Kurakin et al. (2016), we set $\alpha = 1$ and the number of iterations to be $\lfloor \min(\epsilon_{\text{attack}} + 4, 1.25\epsilon_{\text{attack}}) \rfloor$. This method is also called Projected Gradient Descent (PGD) in Madry et al. (2017).

**DeepFool** DeepFool (Moosavi-Dezfooli et al., 2016) works by iteratively linearizing the decision boundary and finding the closest adversarial examples with geometric formulas. However, compared to FGSM and BIM, this method is much slower in practice. We clip the resulting image so that its perturbation is no larger than $\epsilon_{\text{attack}}$.

**Carlini-Wagner (CW)** Carlini & Wagner (2017b) proposed an efficient optimization objective for iteratively finding the adversarial examples with the smallest perturbations. As with DeepFool, we clip the output image to make sure the perturbations are limited by $\epsilon_{\text{attack}}$.

### 2.2 DEFENSE METHODS

Current defense methods generally fall into two classes. They either (1) change the network architecture or training procedure to make it more robust, or (2) modify adversarial examples to reduce their harm. In this paper, we take the following defense methods into comparison.

**Adversarial training** This defense works by generating adversarial examples on-the-fly during training and including them into the training set. FGSM adversarial examples are the most commonly used ones for adversarial training, since they are fast to generate and easy to train. Although training with higher-order adversarial examples (*e.g.*, BIM) has witnessed some success in small datasets (Madry et al., 2017), other work has reported failure in larger ones (Kurakin et al., 2016). We consider both variants in our work.

**Label smoothing**   In contrast to adversarial training, label smoothing (Warde-Farley & Goodfellow, 2016) is agnostic to the attack method. It converts one-hot labels to soft targets, where the correct class has value $1 - \epsilon$ while the other (wrong) classes have value $\epsilon/(N-1)$. Here $\epsilon$ is a small constant and $N$ is the number of classes. When the classifier is re-trained on these soft targets rather than the one-hot labels it is significantly more robust to adversarial examples. This method was originally devised to achieve a similar effect as *defensive distillation* (Papernot et al., 2016c), and their performance is comparable. We didn't compare to defensive distillation since it is more computationally expensive.

**Feature squeezing**   Feature squeezing (Xu et al., 2017a) is both attack-agnostic and model-agnostic. Given any input image, it first reduces the color range from $[0, 255]$ to a smaller value, and then smooths the image with a median filter. The resulting image is then passed to a classifier for predictions. Since this technique does not depend on attacking methods and classifiers, it can be combined with other defensive methods such as adversarial training, similar to PixelDefend.

### 2.3   EXPERIMENT METHODOLOGIES

**Datasets**   Two datasets are used in our experiments: Fashion MNIST (Xiao et al., 2017) and CIFAR-10 (Krizhevsky et al.). Fashion MNIST was designed as a more difficult, but drop-in replacement for MNIST (LeCun et al., 1998). Thus it shares all of MNIST's characteristics, *i.e.*, $60,000$ training examples and $10,000$ test examples where each example is a $28 \times 28$ gray-scale image associated with a label from 1 of 10 classes. CIFAR-10 is another dataset that is also broadly used for image classification tasks. It consists of $60,000$ examples, where $50,000$ are used for training and $10,000$ for testing, and each sample is a $32 \times 32$ color image associated with 1 of 10 classes.

**Models**   We examine two state-of-the-art deep neural network image classifiers: ResNet (He et al., 2016) and VGG (Simonyan & Zisserman, 2014). The architectures are described in Appendix C.

**PixelCNN**   The PixelCNN (van den Oord et al., 2016b; Salimans et al., 2017) is a generative model with tractable likelihood especially designed for images. The model defines the joint distribution over all pixels by factorizing it into a product of conditional distributions.

$$p_{\text{CNN}}(\mathbf{X}) = \prod_i p_{\text{CNN}}(x_i | x_{1:(i-1)}).$$

The pixel dependencies are in raster scan order (row by row and column by column within each row). We train the PixelCNN model for each dataset using only clean (not perturbed) image samples. In Appendix D, we provide clean sample images from the datasets as well as generated image samples from PixelCNN (see Figure 8 and Figure 9).

As a convenient representation of $p_{\text{CNN}}(\mathbf{X})$ for images, we also use the concept of *bits per dimension*, which is defined as $\text{BPD}(\mathbf{X}) \triangleq -\log p_{\text{CNN}}(\mathbf{X})/(I \times J \times K \times \log 2)$ for an image of resolution $I \times J$ and $K$ channels.

## 3   DETECTING ADVERSARIAL EXAMPLES

Adversarial images are defined with respect to a specific classifier. Intuitively, a maliciously perturbed image that causes one network to give a highly confident incorrect prediction might not fool another network. However, recent work (Papernot et al., 2016a; Liu et al., 2016; Tramèr et al., 2017) has shown that adversarial images can transfer across different classifiers. This indicates that there are some intrinsic properties of adversarial examples that are independent of classifiers.

One possibility is that, compared to normal training and test images, adversarial examples have much lower probability densities under the image distribution. As a result, classifiers do not have enough training instances to get familiarized with this part of the input space. The resulting prediction task suffers from covariate shift, and since all of the classifiers are trained on the same dataset, this covariate shift will affect all of them similarly and will likely lead to misclassifications.

To empirically verify this hypothesis, we train a PixelCNN model on the CIFAR-10 (Krizhevsky & Hinton, 2009) dataset and use its log-likelihood as an approximation to the true underlying probability density. The adversarial examples are generated with respect to a ResNet (He et al., 2016), which

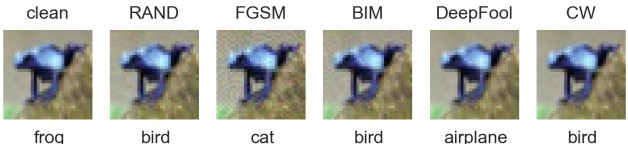

Figure 1: An image sampled from the CIFAR-10 test dataset and various adversarial examples generated from it. The text above shows the attacking method while the text below shows the predicted label of the ResNet.

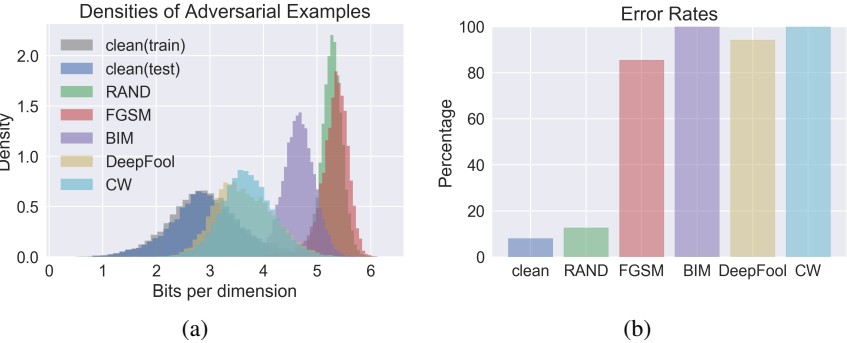

Figure 2: (a) Likelihoods of different perturbed images with $\epsilon_{\text{attack}} = 8$. (b) Test errors of a ResNet on different adversarial examples.

gets 92% accuracy on the test images. We generate adversarial examples from RAND, FGSM, BIM, DeepFool and CW methods with $\epsilon_{\text{attack}} = 8$. Note that as shown in Figure 1, the resulting adversarial perterbations are barely perceptible to humans.

However, the distribution of log-likelihoods show considerable difference between perturbed images and clean images. As summarized in Figure 2, even a 3% perturbation can lead to systematic decrease of log-likelihoods. Note that the PixelCNN model has no information about the attacking methods for producing those adversarial examples, and no information about the ResNet model either.

We can see from Figure 3(b) that random perturbations also push the images outside of the training distribution, even though they do not have the same adverse effect on accuracy. We believe this is due to an inductive bias that is shared by many neural network models but not inherent to all models, as discussed further in Appendix A.

Besides qualitative analysis, the log-likelihoods from PixelCNN also provide a quantitative measure for detecting adversarial examples. Combined with permutation test (Efron & Tibshirani, 1994), we can provide a uncertainty value for each input about whether it comes from the training distribution or not. Specifically, let the input $\mathbf{X}' \overset{\text{i.i.d.}}{\sim} q(\mathbf{X})$ and training images $\mathbf{X}_1, \cdots, \mathbf{X}_N \overset{\text{i.i.d.}}{\sim} p(\mathbf{X})$. The null hypothesis is $H_0 : p(\mathbf{X}) = q(\mathbf{X})$ while the alternative is $H_1 : p(\mathbf{X}) \neq q(\mathbf{X})$. We first compute the probabilities give by a PixelCNN for $\mathbf{X}'$ and $\mathbf{X}_1, \cdots, \mathbf{X}_N$, then use the rank of $p_{\text{CNN}}(\mathbf{X}')$ in $\{p_{\text{CNN}}(\mathbf{X}_1), \cdots, p_{\text{CNN}}(\mathbf{X}_N)\}$ as our test statistic:

$$T = T(\mathbf{X}'; \mathbf{X}_1, \cdots, \mathbf{X}_N) \triangleq \sum_{i=1}^{N} \mathbb{I}[p_{\text{CNN}}(\mathbf{X}_i) \leq p_{\text{CNN}}(\mathbf{X}')].$$

Here $\mathbb{I}[\cdot]$ is the indicator function, which equals 1 when the condition inside brackets is true and otherwise equals 0. Let $T_i = T(\mathbf{X}_i; \mathbf{X}_1, \cdots, \mathbf{X}_{i-1}, \mathbf{X}', \mathbf{X}_{i+1}, \cdots, \mathbf{X}_N)$. According to the permutation principle, $T_i$ has the same distribution as $T$ under the null hypothesis $H_0$. We can therefore compute the $p$-value exactly by

$$p = \frac{1}{N+1} \left( \sum_{i=1}^{N} \mathbb{I}[T_i \leq T] + 1 \right) = \frac{T+1}{N+1} = \frac{1}{N+1} \left( \sum_{i=1}^{N} \mathbb{I}[p_{\text{CNN}}(\mathbf{X}_i) \leq p_{\text{CNN}}(\mathbf{X}')] + 1 \right).$$

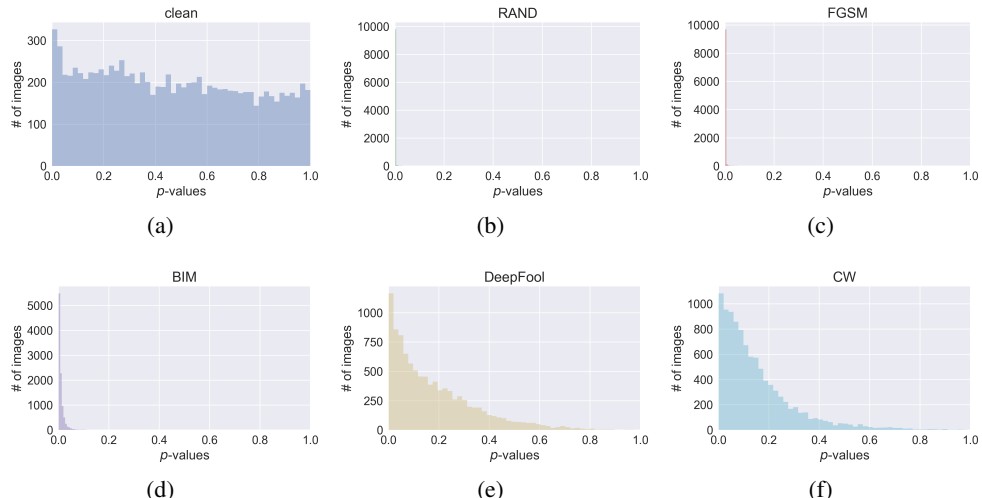

Figure 3: The distribution of $p$-values under the PixelCNN generative model. The inputs are more outside of the training distribution if their $p$-value distribution has a larger deviation from uniform. Here "clean" means clean test images. From definition, the $p$-values of clean training images have a uniform distribution.

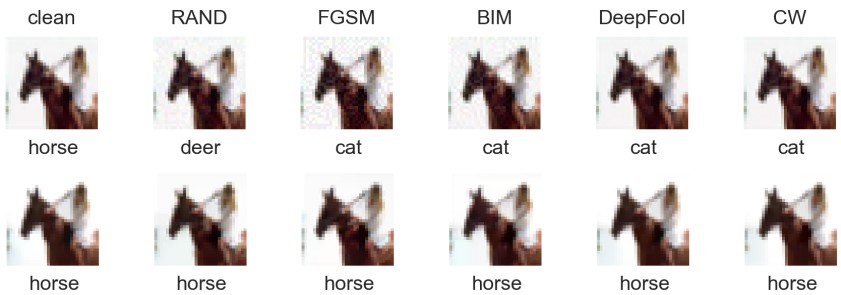

Figure 4: An example of how purification works. The above row shows an image from CIFAR-10 test set and various attacking images generated from it. The bottom row shows corresponding purified images. The text below each image is the predicted label given by our ResNet.

For CIFAR-10, we provide histograms of $p$-values for different adversarial examples in Figure 3 and ROC curves of using $p$-values for detection in Figure 6(a). Note that in the ideal case, the $p$-value distribution of clean test images should be uniform. The method works especially well for attacks producing larger perturbations, such as RAND, FGSM, and BIM. For DeepFool and CW adversarial examples, we can also observe significant deviations from uniform. As shown in Figure 3(a), the $p$-value distribution of test images are almost uniform, indicating good generalization of the PixelCNN model.

## 4    PURIFYING IMAGES WITH PIXELDEFEND

In many circumstances, simply detecting adversarial images is not sufficient. It is often critical to be able to correctly classify images despite such adversarial modifications. In this section we introduce PixelDefend, a specific instance of a new family of defense methods that significantly improves the state-of-the-art performance on advanced attacks, while simultaneously performing well against all other attacks.

---

**Algorithm 1** PixelDefend

---

**Input:** Image $\mathbf{X}$, Defense parameter $\epsilon_{\text{defend}}$, Pre-trained PixelCNN model $p_{\text{CNN}}$
**Output:** Purified Image $\mathbf{X}^*$
  1: $\mathbf{X}^* \leftarrow \mathbf{X}$
  2: **for** each row $i$ **do**
  3:     **for** each column $j$ **do**
  4:         **for** each channel $k$ **do**
  5:             $x \leftarrow \mathbf{X}[i, j, k]$
  6:             Set feasible range $R \leftarrow [\max(x - \epsilon_{\text{defend}}, 0), \min(x + \epsilon_{\text{defend}}, 255)]$
  7:             Compute the 256-way softmax $p_{\text{CNN}}(\mathbf{X}^*)$.
  8:             Update $\mathbf{X}^*[i, j, k] \leftarrow \arg\max_{z \in R} p_{\text{CNN}}[i, j, k, z]$
  9:         **end for**
 10:     **end for**
 11: **end for**

---

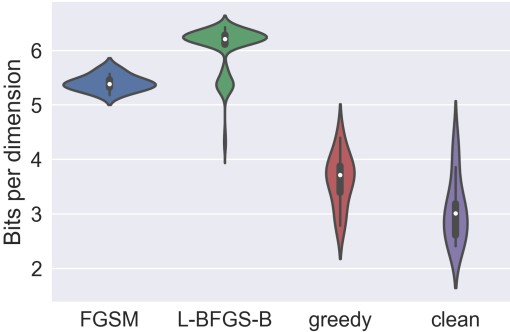

Figure 5: The bits-per-dimension distributions of purified images from FGSM adversarial examples. We tested two purification methods, L-BFGS-B and greedy decoding, the latter of which is used in PixelDefend. A good purification method should give images that have lower bits per dimension compared to FGSM images and ideally similar bits per dimension compared to clean ones.

## 4.1 RETURNING IMAGES TO THE TRAINING DISTRIBUTION

The basic idea behind PixelDefend is to *purify* input images, by making small changes to them in order to move them back towards the training distribution, *i.e.*, move the images towards a high-probability region. We then classify the purified image using any existing classifier. As the example in Figure 4 shows, the purified images can usually be classified correctly.

Formally, we have training image distribution $p(\mathbf{X})$, and input image $\mathbf{X}$ of resolution $I \times J$ with $\mathbf{X}[i, j, k]$ the pixel at location $(i, j)$ and channel $k \in \{1, \cdots, C\}$. We wish to find an image $\mathbf{X}^*$ that maximizes $p(\mathbf{X})$ subject to the constraint that $\mathbf{X}^*$ is within the $\epsilon_{\text{defend}}$-ball of $\mathbf{X}$:

$$\max_{\mathbf{X}^*} p(\mathbf{X}^*)$$
$$\text{s.t.} \quad \|\mathbf{X}^* - \mathbf{X}\|_\infty \leq \epsilon_{\text{defend}}.$$

Here $\epsilon_{\text{defend}}$ reflects a trade-off, since large $\epsilon_{\text{defend}}$ may change the meaning of $\mathbf{X}$ while small $\epsilon_{\text{defend}}$ may not be sufficient for returning $\mathbf{X}$ to the correct distribution. In practice, we choose $\epsilon_{\text{defend}}$ to be some value that overestimates $\epsilon_{\text{attack}}$ but still keeps high accuracies on clean images. As in Section 3, we approximate $p(\mathbf{X})$ with the PixelCNN distribution $p_{\text{CNN}}(\mathbf{X})$, which is trained on the same training set as the classifier.

However, exact constrained optimization of $p_{\text{CNN}}(\mathbf{X})$ is computationally intractable. Surprisingly, even gradient-based optimization faces great difficulty on that problem. We found that one advanced methods in gradient-based constrained optimization, L-BFGS-B (Byrd et al., 1995) (we use the `scipy` implementation based on Zhu et al. (1997)), actually decreases $p_{\text{CNN}}(\mathbf{X})$ for most random initializations within the $\epsilon_{\text{defend}}$-ball.

For efficient optimization, we instead use a greedy technique described in Algorithm 1, which is similar to the greedy decoding process typically used in sequence-to-sequence models (Sutskever et al., 2014). The method is similar to generating images from PixelCNN, with the additional constraint that the generated image should be within an $\epsilon_{\text{defend}}$-ball of a perturbed image. As an autoregressive model, PixelCNN is slow in image generation. Nonetheless, by caching redundant calculation, Ramachandran et al. (2017) proposes a very fast generation algorithm for PixelCNN. In our experiments, adoption of Ramachandran et al. (2017)'s method greatly increases the speed of PixelDefend. For CIFAR-10 images, PixelDefend on average processes 3.6 images per second on one NVIDIA TITAN Xp GPU.

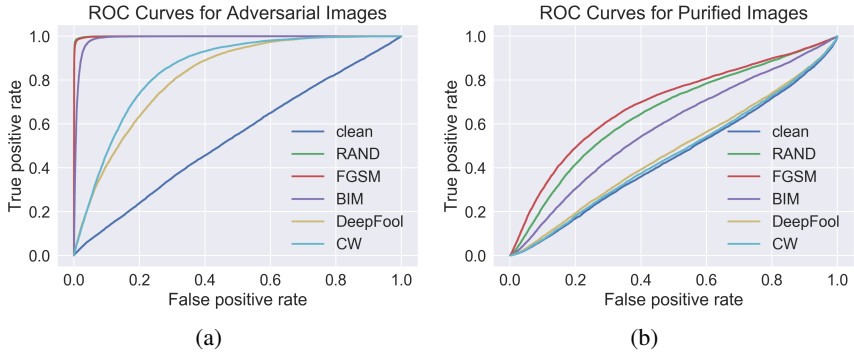

Figure 6: ROC curves showing the efficacy of using $p$-values as scores to detect adversarial examples. For computing the ROC, we assign negative labels to training images and positive labels to adversarial images (or clean test images). (a) Original adversarial examples. (b) Purified adversarial examples after PixelDefend.

To show the effectiveness of this greedy method compared to L-BFGS-B, we take the first 10 images from CIFAR-10 test set, attack them by FGSM with $\epsilon_{\text{attack}} = 8$, and purify them with L-BFGS-B and PixelDefend respectively. We used random start points for L-BFGS-B and repeated 100 times for each image. As depicted in Figure 5, most L-BFGS-B attempts failed at minimizing the bits per dimension of FGSM adversarial examples. Because of the rugged gradient landscape of PixelCNN, L-BFGS-B even results in images that have lower probabilities. In contrast, PixelDefend works much better in increasing the probabilities of purified images, although their probabilities are still lower compared to clean ones.

In Figure 6 and Figure 7, we empirically show that after PixelDefend, purified images are more likely to be drawn from the training distribution. Specifically, Figure 6 shows that the detecting power of $p$-values greatly decreases for purified images. For DeepFool and CW examples, purification makes them barely distinguishable from normal samples of the data distribution. This is also manifested by Figure 7, as the $p$-value distributions of purified examples are closer to uniform. Visually, purified images indeed look much cleaner than adversarially perturbed ones. In Appendix E, we provide sampled purified images from Fashion MNIST and CIFAR-10.

## 4.2 Adaptive PixelDefend

One concern with the approach of purifying images is what happens when we purify a clean image. More generally, we will never know $\epsilon_{\text{attack}}$ and if we set $\epsilon_{\text{defend}}$ too large for a given attack, then we will modify all images to become the mode image, which would mostly result in misclassifications. One way to avoid this problem is to tune $\epsilon_{\text{defend}}$ adaptively based on the probability of the input image under the generative model. In this way, images that already have high probability under the training distribution would have a very low $\epsilon_{\text{defend}}$ preventing significant modification, while low probability images would have a high $\epsilon_{\text{defend}}$ thus allowing significant modifications. We implemented a very simple thresholding version of this, which sets $\epsilon_{\text{defend}}$ to zero if the input image probability is below a threshold value, and otherwise leaves it fixed at a manually chosen setting. In practice, we set this threshold based on knowledge of the set of possible attacks, so strictly speaking, the adaptive version of our technique is no longer attack-agnostic.

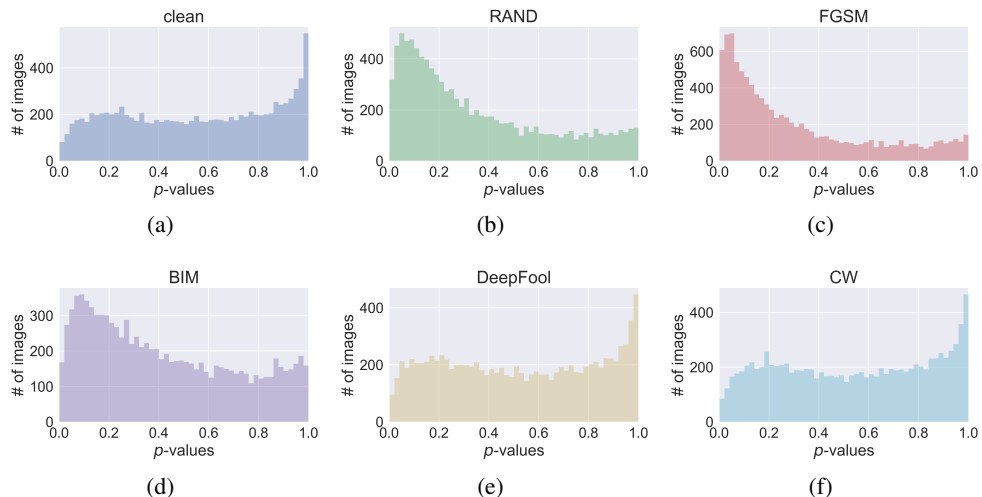

Figure 7: The distributions of $p$-values under the PixelCNN model after PixelDefend purification.

Table 1: **Fashion MNIST** ($\epsilon_{\text{attack}} = 8/25$, $\epsilon_{\text{defend}} = 32$)

| NETWORK | TRAINING TECHNIQUE | CLEAN | RAND | FGSM | BIM | DEEP FOOL | CW | STRONGEST ATTACK |
|---|---|---|---|---|---|---|---|---|
| ResNet | Normal | **93**/**93** | 89/71 | 38/24 | 00/00 | 06/06 | 20/01 | 00/00 |
| VGG | Normal | 92/92 | 91/87 | 73/58 | 36/08 | 49/14 | 43/23 | 36/08 |
| ResNet | Adversarial FGSM | **93**/**93** | **92**/89 | 85/85 | 51/00 | 63/07 | 67/21 | 51/00 |
|  | Adversarial BIM | 92/91 | **92**/**91** | 84/79 | 76/63 | 82/72 | 81/70 | 76/63 |
|  | Label Smoothing | **93**/**93** | 91/76 | 73/45 | 16/00 | 29/06 | 33/14 | 16/00 |
|  | Feature Squeezing | 84/84 | 84/70 | 70/28 | 56/25 | 83/83 | 83/83 | 56/25 |
|  | Adversarial FGSM + Feature Squeezing | 88/88 | 87/82 | 80/77 | 70/46 | 86/82 | 84/85 | 70/46 |
| ResNet | Normal + *PixelDefend* | 88/88 | 88/89 | 85/74 | 83/76 | 87/87 | 87/87 | 83/74 |
| VGG | Normal + *PixelDefend* | 89/89 | 89/89 | 87/82 | 85/83 | 88/88 | 88/88 | 85/82 |
| ResNet | Adversarial FGSM + *PixelDefend* | 90/89 | 91/90 | **88**/82 | **85**/76 | **90**/88 | **89**/**88** | **85**/76 |
|  | Adversarial FGSM +*Adaptive PixelDefend* | 91/91 | 91/**91** | **88**/**88** | **85**/84 | 89/**90** | **89**/84 | **85**/**84** |

## 4.3 PIXELDEFEND RESULTS

We carried out a comprehensive set of experiments to test various defenses versus attacks. Detailed information on experimental settings is provided in Appendix B. All experimental results are summarized in Tab. 1 and Tab. 2. In the upper part of the tables, we show how the various baseline defenses fare against each of the attacks, while in the lower part of the tables we show how our PixelDefend technique works. Each table cell contains accuracies on adversarial examples generated with different $\epsilon_{\text{attack}}$. More specifically, for Fashion MNIST (Tab. 1), we tried $\epsilon_{\text{attack}} = 8$ and 25. The cells in Tab. 1 is formated as $x/y$, where $x$ denotes the accuracy (%) on images attacked with $\epsilon_{\text{attack}} = 8$, while $y$ denotes the accuracy when $\epsilon_{\text{attack}} = 25$. For CIFAR-10 (Tab. 2), we tried $\epsilon_{\text{attack}} = 2, 8$, and 16, and the cells are formated in a similar way. We use the same $\epsilon_{\text{defend}}$ for different $\epsilon_{\text{attack}}$'s to show that PixelDefend is insensitive to $\epsilon_{\text{attack}}$.

From the tables we observe that adversarial training successfully defends against the basic FGSM attack, but cannot defend against the more advanced ones. This is expected, as training on simple adversarial examples does not guarantee robustness to more complicated attacking techniques. Consistent with Madry et al. (2017), adversarial training with BIM examples is more successful at preventing a wider spectrum of attacks. For example, it improves the accuracy on strongest attack from 2% to 32% on CIFAR-10 when $\epsilon_{\text{attack}} = 8$. But the numbers are still not ideal even with respect

Table 2: **CIFAR-10** ($\epsilon_{\text{attack}} = 2/8/16$, $\epsilon_{\text{defend}} = 16$)

| NETWORK | TRAINING TECHNIQUE | CLEAN | RAND | FGSM | BIM | DEEP FOOL | CW | STRONGEST ATTACK |
|---------|-----------|-------|------|------|-----|-----------|-----|---------|
| ResNet | Normal | **92/92/92** | **92**/87/76 | 33/15/11 | 10/00/00 | 12/06/06 | 07/00/00 | 07/00/00 |
| VGG | Normal | 89/89/89 | 89/88/80 | 60/46/30 | 44/02/00 | 57/25/11 | 37/00/00 | 37/00/00 |
| ResNet | Adversarial FGSM | 91/91/91 | 90/**88**/84 | **88/91/91** | 24/07/00 | 45/00/00 | 20/00/07 | 20/00/00 |
| | Adversarial BIM | 87/87/87 | 87/87/86 | 80/52/34 | 74/32/06 | 79/48/25 | 76/42/08 | 74/32/06 |
| | Label Smoothing | **92/92/92** | 91/**88**/77 | 73/54/28 | 59/08/01 | 56/20/10 | 30/02/02 | 30/02/01 |
| | Feature Squeezing | 84/84/84 | 83/82/76 | 31/20/18 | 13/00/00 | 75/75/75 | 78/78/78 | 13/00/00 |
| | Adversarial FGSM + Feature Squeezing | 86/86/86 | 85/84/81 | 73/67/55 | 55/02/00 | **85/85/85** | 83/83/83 | 55/02/00 |
| ResNet | Normal + *PixelDefend* | 85/85/88 | 82/83/84 | 73/46/24 | 71/46/25 | 80/80/80 | 78/78/78 | 71/46/24 |
| VGG | Normal + *PixelDefend* | 82/82/82 | 82/82/84 | 80/62/52 | 80/61/48 | 81/76/76 | 81/79/79 | 80/61/48 |
| ResNet | Adversarial FGSM + *PixelDefend* | 88/88/86 | 86/86/**87** | 81/68/67 | **81**/69/**56** | **85/85/85** | 84/84/84 | **81**/69/**56** |
| | Adversarial FGSM + *Adaptive PixelDefend* | 90/90/90 | 86/87/**87** | 81/70/67 | **81/70/56** | 82/81/82 | 81/80/81 | **81/70/56** |

to BIM attack itself. As in Tab. 2, it only gets 6% on BIM and 8% on CW when $\epsilon_{\text{attack}} = 16$. We also observe that label smoothing, which learns smoothed predictions so that the gradient $\nabla_{\mathbf{X}} L(\mathbf{X}, y)$ becomes very small, is only effective against simple FGSM attack. Model-agnostic methods, such as feature squeezing, can be combined with other defenses for strengthened performance. We observe that combining it with adversarial training indeed makes it more robust. Actually, Tab. 1 and Tab. 2 show that feature squeezing combined with adversarial training dominates using feature squeezing along in all settings. It also gets good performance on DeepFool and CW attacks. However, for iterative attacks with larger perturbations, *i.e.*, BIM, feature squeezing performs poorly. On CIFAR-10, it only gets 2% and 0% accuracy on BIM with $\epsilon_{\text{attack}} = 8$ and 16 respectively.

PixelDefend, our model-agnostic and attack-agnostic method, performs well on different classifiers (ResNet and VGG) and different attacks *without modification*. In addition, we can see that augmenting basic adversarial training with PixelDefend can sometimes double the accuracies. We hypothesize that the purified images from PixelDefend are still not perfect, and adversarially trained networks have more toleration for perturbations. This also corroborates the plausibility and benefit of combining PixelDefend with other defenses.

Furthermore, PixelDefend can simultaneously obtain accuracy above 70% for all other attacking techniques, while ensuring that performance on clean images only declines slightly. Models with PixelDefend consistently outperform other methods with respect to the strongest attack. On Fashion MNIST, PixelDefend methods improve the accuracy on strongest attack from 76% to 85% and 63% to 84%. On CIFAR-10, the improvements are even more significant, *i.e.*, from 74% to 81%, 32% to 70% and 6% to 56%, for $\epsilon_{\text{attack}} = 2$, 8, and 16 respectively. In a security-critical scenario, the weakest part of a system determines the overall reliability. Therefore, the outstanding performance of PixelDefend on the strongest attack makes it a valuable and useful addition for improving AI security.

## 4.4 END-TO-END ATTACK OF PIXELDEFEND

A natural question that arises is whether we can generate a new class of adversarial examples targeted specifically at the combined PixelDefend architecture of first purifying the image and then using an existing classifier to predict the label of the purified image. We have three pieces of empirical evidence to believe that such adversarial examples are hard to find in general. First, we attempted to apply the iterative BIM attack to an end-to-end differentiable version of PixelDefend generated by unrolling the PixelCNN purification process. However we found the resulting network was too deep and led to problems with vanishing gradients (Bengio et al., 1994), resulting in adversarial images that were identical to the original images. Moreover, attacking the whole system is very time consuming. Empirically, it took about 10 hours to generate 100 attacking images with one TITAN Xp GPU which failed to fool PixelDefend. Secondly, we found the optimization problem in Eq. (4.1) was not amenable to gradient descent, as indicated in Figure 5. This makes gradient-based attacks

especially difficult. Last but not least, the generative model and classifier are trained separately and have independent parameters. Therefore, the perturbation direction that leads to higher probability images has a smaller correlation with the perturbation direction that results in misclassification. Accordingly, it is harder to find adversarial examples that can fool both of them together. However, we will open source our codes and look forward to any possible attack from the community.

## 5 RELATED WORK

Most recent work on detecting adversarial examples focuses on adding an outlier class detection module to the classifier, such as Grosse et al. (2017), Gong et al. (2017) and Metzen et al. (2017). Those methods require the classification model to be changed, and are thus not model-agnostic. Feinman et al. (2017) also presents a detection method based on kernel density estimation and Bayesian neural network uncertainty. However, Carlini & Wagner (2017a) shows that all those methods can be bypassed.

Grosse et al. (2017) also studied the distribution of adversarial examples from a statistical testing perspective. They reported the same discovery that adversarial examples are outside of the training distribution. However, our work is different from theirs in several important aspects. First, the kernel-based two-sample test used in their paper needs a large number of suspicious inputs, while our method only requires one data point. Second, they mainly tested on first-order methods such as FGSM and JSMA (Papernot et al., 2016b). We show the efficacy of PixelCNN on a wider range of attacking methods (see Figure 3), including both first-order and iterative methods. Third, we further demonstrate that random perturbed inputs are also outside of the training distribution.

Some other work has focused on modifying the classifier architecture to increase its robustness, *e.g.*, Gu & Rigazio (2014), Cisse et al. (2017) and Nayebi & Ganguli (2017). Although they have witnessed some success, such modifications of models might limit their representative power and are also not model-agnostic.

Our basic idea of moving points to higher-density regions is also present in other machine learning methods not specifically designed for handling adversarial data; for example, the *manifold denoising method* of Hein & Maier (2007), the *direct density gradient estimation* of Sasaki et al. (2014), and the *denoising autoencoders* of Vincent et al. (2008) all move data points from low to high-density regions. In the future some of these methods could be adapted to amortize the purification process directly, that is, to learn a *purification network*.

## 6 CONCLUSION

In this work, we discovered that state-of-the-art neural density models, *e.g.*, PixelCNN, can detect small perturbations with high sensitivity. This sensitivity broadly exists for a large number of perturbations generated with different methods. An interesting fact is that PixelCNN is only sensitive in one direction—it is relatively easy to detect perturbations that lead to lower probabilities rather than higher probabilities.

Based on the sensitivity of PixelCNN, we utilized statistical hypothesis testing to verify that adversarial examples lie outside of the training distribution. With the permutation test, we give exact $p$-values which can be used as a uncertainty measure for detecting outlier perturbations.

Furthermore, we make use of the sensitivity of generative models to explore the idea of purifying adversarial examples. We propose the PixelDefend algorithm, and experimentally show that returning adversarial examples to high probability regions of the training distribution can significantly decrease their damage to classifiers. Different from many other defensive techniques, PixelDefend is model-agnostic and attack-agnostic, which means it can be combined with other defenses to improve robustness without modifying the classification model. As a result PixelDefend is a practical and effective defense against adversarial inputs.

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

## APPENDIX A    ON RANDOM PERTURBATIONS

One may observe from Figure 3(b) that random perturbations have very low $p$-values, and thus also live outside of the high density area. Although many classifiers are robust to random noise, it is not a property granted by the dataset. The fact is that robustness to random noise could be from model inductive bias, and there exist classifiers which have high generalization performance on clean images, but can be attacked by small random perturbations.

It is easy to construct a concrete classifier that are susceptible to random perturbations. Our ResNet on CIFAR-10 gets 92.0% accuracy on the test set and 87.3% on randomly perturbed test images with $\epsilon_{\text{attack}} = 8$. According to our PixelCNN, 175 of 10000 test images have a bits per dimension (BPD) larger than 4.5, while the number for random images is 9874. Therefore, we can define a new classifier

$$\text{ResNet'}(\mathbf{X}) \triangleq \begin{cases} \text{ResNet}(\mathbf{X}), & \text{BPD}(\mathbf{X}) < 4.5 \\ \text{random label}, & \text{BPD}(\mathbf{X}) \geq 4.5 \end{cases},$$

which will get roughly $92\% \times 9825/10000 + 10\% \times 175/10000 \approx 90.6\%$ accuracy on the test set, while only $87.3\% \times 126/10000 + 10\% \times 9874/10000 \approx 11.0\%$ accuracy on the randomly perturbed images. This classifier has comparable generalization performance to the original ResNet, but will give incorrect labels to most randomly perturbed images.

## APPENDIX B    EXPERIMENTAL SETTINGS

**Adversarial Training**    We have tested adversarial training with both FGSM and BIM examples. During training, we take special care of the label leaking problem as noted in Kurakin et al. (2016)— we use the predicted labels of the model to generate adversarial examples, instead of using the true labels. This prevents the adversarially trained network to perform better on adversarial examples than clean images by simply retrieving ground-truth labels. Following Kurakin et al. (2016), we also sample $\epsilon_{\text{attack}}$ from a truncated Gaussian distribution for generating FGSM or BIM adversarial examples, so that the adversarially trained network won't overfit to any specific $\epsilon_{\text{attack}}$. This is different from Madry et al. (2017), where the authors train and test with the same $\epsilon_{\text{attack}}$.

For Fashion MNIST experiments, we randomly sample $\epsilon_{\text{attack}}$ from $\mathcal{N}(0, \delta)$, take the absolute value and truncate it to $[0, 2\delta]$, where $\delta = 8$ or $25$. For CIFAR-10 experiments, we follow the same procedure but fix $\delta = 8$.

**Feature Squeezing**    For implementing the feature squeezing defense, we reduce the number of colors to 8 on Fashion MNIST, and use 32 colors for CIFAR-10. The numbers are chosen to make sure color reduction will not lead to significant deterioration of image quality. After color depth reduction, we apply a $2 \times 2$ median filter with reflective paddings, since it is reported in Xu et al. (2017b) to be most effective for preventing CW attacks.

**Models**    We use ResNet (62-layer) and VGG (16-layer) as classifiers. In our experiments, normally trained networks have the same architectures as adversarially trained networks. Since the images of Fashion MNIST contain roughly one quarter values of those of CIFAR-10, we use a smaller network for classifying Fashion MNIST. More specifically, we reduce the number of feature maps for Fashion MNIST to 1/4 while keeping the same depths. In practive, VGG is more robust than ResNet due to using of dropout layers. The network architecture details are described in Appendix C. For the PixelCNN generative model, we adopted the implementation of PixelCNN++ (Salimans et al., 2017), but modified the output from mixture of logistic distributions to softmax. The feature maps are also reduced to 1/4 for training PixelCNN on Fashion MNIST.

**Adaptive Threshold**    We chose the adaptive threshold discussed in Section 4.2 using validation data. We set the threshold at the lowest value which did not decrease the performance of the strongest adversary. For Fashion MNIST, the threshold of bits per dimension was set to 1.8, and for CIFAR-10 the number was 3.2. As a reference, the mean value of bits per dimension for Fashion MNIST test images is 2.7 and for CIFAR-10 is 3.0. However, we admit that using a validation set to choose the best threshold makes the adaptive version of PixelDefend not strictly attack-agnostic.

## APPENDIX C   IMAGE CLASSIFIER ARCHITECTURES*

### C.1   RESNET CLASSIFIER FOR CIFAR-10 & FASHION MNIST

| NAME | CONFIGURATION | |
|------|---------------|--|
| Initial Layer | conv (filter size: $3 \times 3$, feature maps: 16 (4), stride size: $1 \times 1$) | |
| Residual Block 1 | batch normalization & leaky relu
conv (filter size: $3 \times 3$, feature maps: 16 (4), stride size: $1 \times 1$)
batch normalization & leaky relu
conv (filter size: $3 \times 3$, feature maps: 16 (4), stride size: $1 \times 1$)
residual addition | $\times 10$ times |
| Residual Block 2 | batch normalization & leaky relu
conv (filter size: $3 \times 3$, feature maps: 32 (8), stride size: $2 \times 2$)
batch normalization & leaky relu
conv (filter size: $3 \times 3$, feature maps: 32 (8), stride size: $1 \times 1$)
average pooling & padding & residual addition | |
| | batch normalization & leaky relu
conv (filter size: $3 \times 3$, feature maps: 32 (8), stride size: $1 \times 1$)
batch normalization & leaky relu
conv (filter size: $3 \times 3$, feature maps: 32 (8), stride size: $1 \times 1$)
residual addition | $\times 9$ times |
| Residual Block 3 | batch normalization & leaky relu
conv (filter size: $3 \times 3$, feature maps: 64 (16), stride size: $2 \times 2$)
batch normalization & leaky relu
conv (filter size: $3 \times 3$, feature maps: 64 (16), stride size: $1 \times 1$)
average pooling & padding & residual addition | |
| | batch normalization & leaky relu
conv (filter size: $3 \times 3$, feature maps: 64 (16), stride size: $1 \times 1$)
batch normalization & leaky relu
conv (filter size: $3 \times 3$, feature maps: 64 (16), stride size: $1 \times 1$)
residual addition | $\times 9$ times |
| Pooling Layer | batch normalization & leaky relu & average pooling | |
| Output Layer | fc_10 & softmax | |

### C.2   VGG CLASSIFIER FOR CIFAR-10 & FASHION MNIST

| NAME | CONFIGURATION | |
|------|---------------|--|
| Feature Block 1 | conv (filter size: $3 \times 3$, feature maps: 16 (4), stride size: $1 \times 1$)
batch normalization & relu | $\times 2$ times |
| | max pooling (stride size: $2 \times 2$) | |
| Feature Block 2 | conv (filter size: $3 \times 3$, feature maps: 128 (32), stride size: $1 \times 1$)
batch normalization & relu | $\times 2$ times |
| | max pooling (stride size: $2 \times 2$) | |
| Feature Block 3 | conv (filter size: $3 \times 3$, feature maps: 512 (128), stride size: $1 \times 1$)
batch normalization & relu | $\times 3$ times |
| | max pooling (stride size: $2 \times 2$) | |
| Feature Block 4 | conv (filter size: $3 \times 3$, feature maps: 512 (128), stride size: $1 \times 1$)
batch normalization & relu | $\times 3$ times |
| | max pooling (stride size: $2 \times 2$) & flatten | |
| Classifier Block | dropout & fc_512 (128) & relu
dropout & fc_10 & softmax | |

---

*The same architecture is used for both CIFAR-10 and Fashion MNIST, but different numbers of feature maps are used. The number of feature maps in parentheses is for Fashion MNIST.

# APPENDIX D    SAMPLED IMAGES FROM PIXELCNN

## D.1    FASHION MNIST

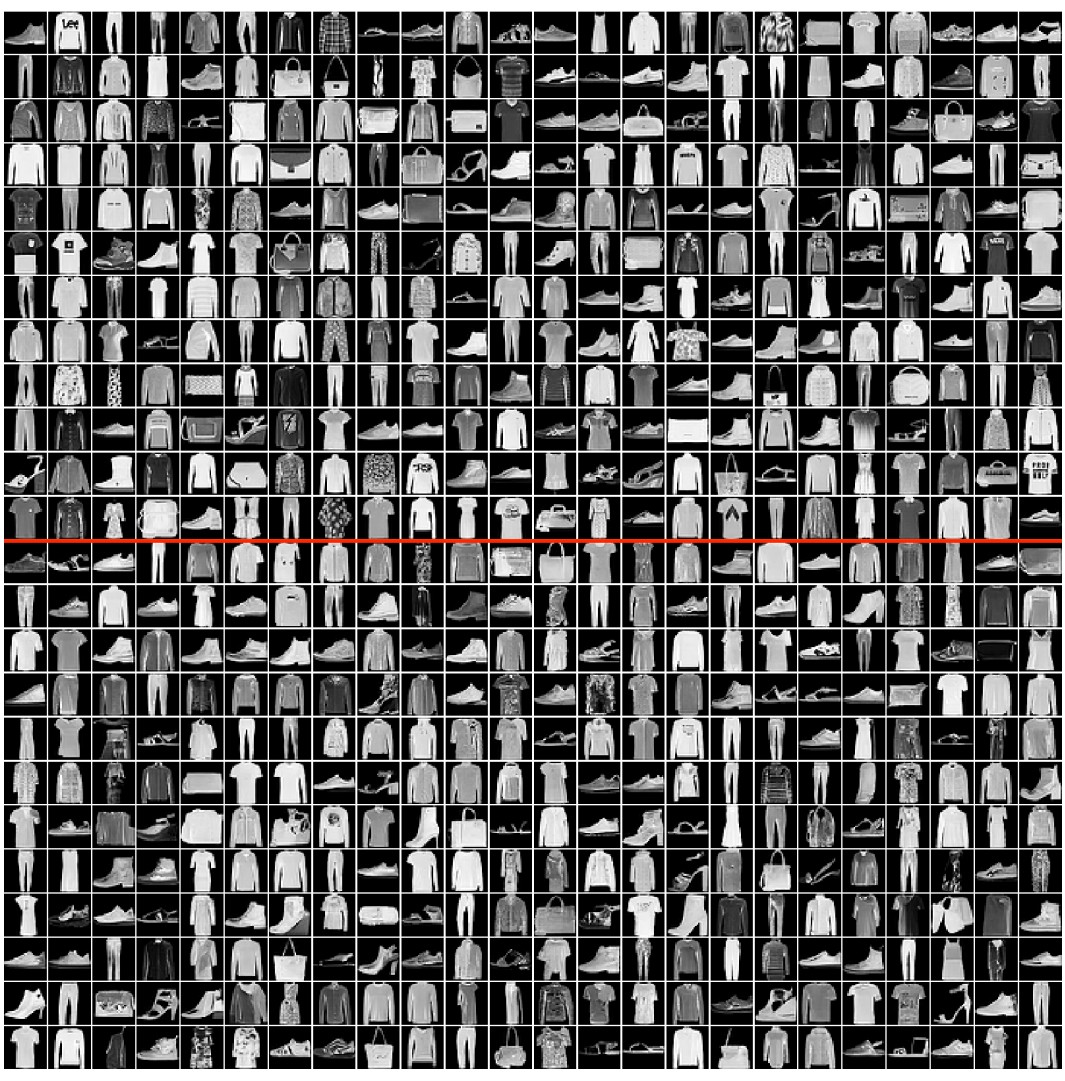

Figure 8: True and generated images from Fashion MNIST. The upper part shows true images sampled from the dataset while the bottom shows generated images from PixelCNN.

## D.2 CIFAR-10

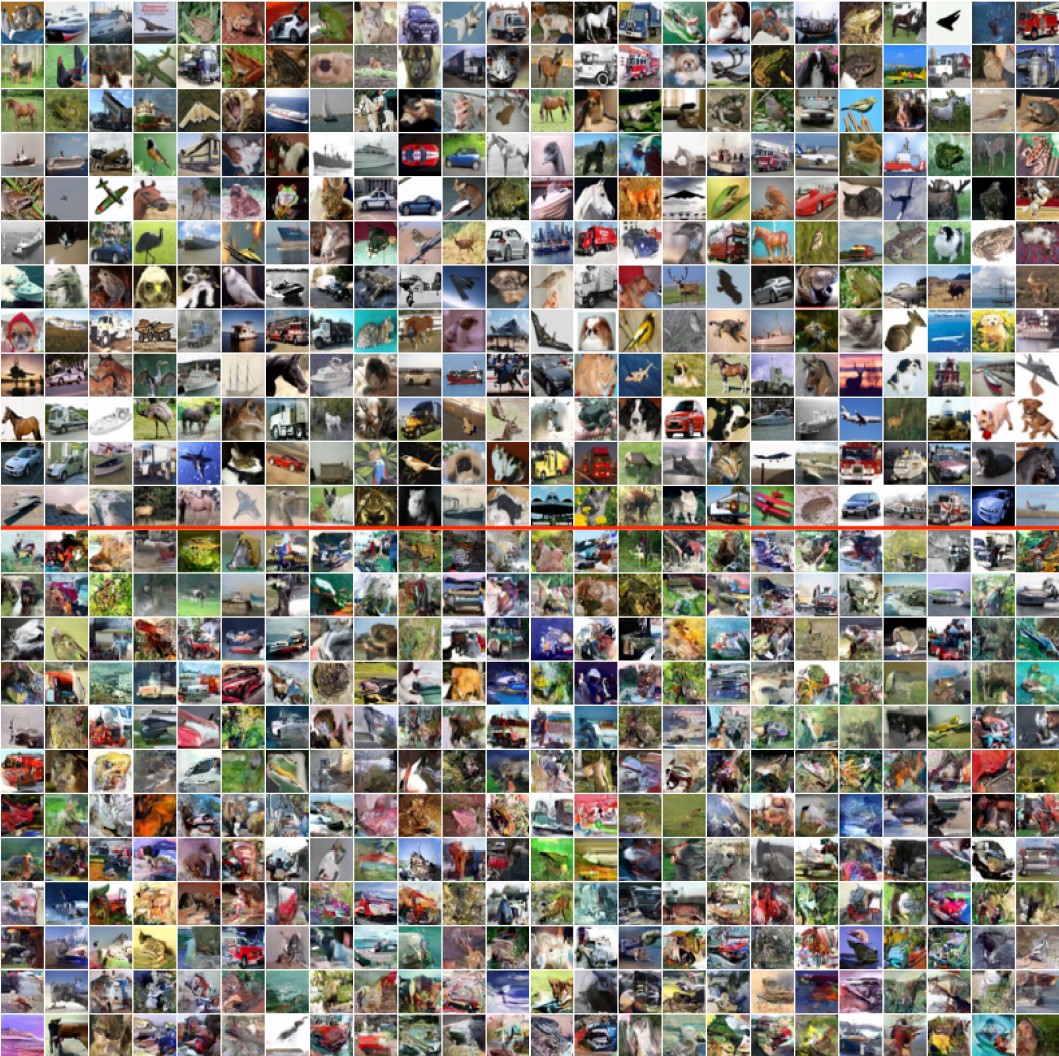

Figure 9: True and generated images from CIFAR-10. The upper part shows true images sampled from the dataset while the bottom part shows generated images from PixelCNN.

## APPENDIX E    SAMPLED PURIFIED IMAGES FROM PIXELDEFEND

### E.1    FASHION MNIST

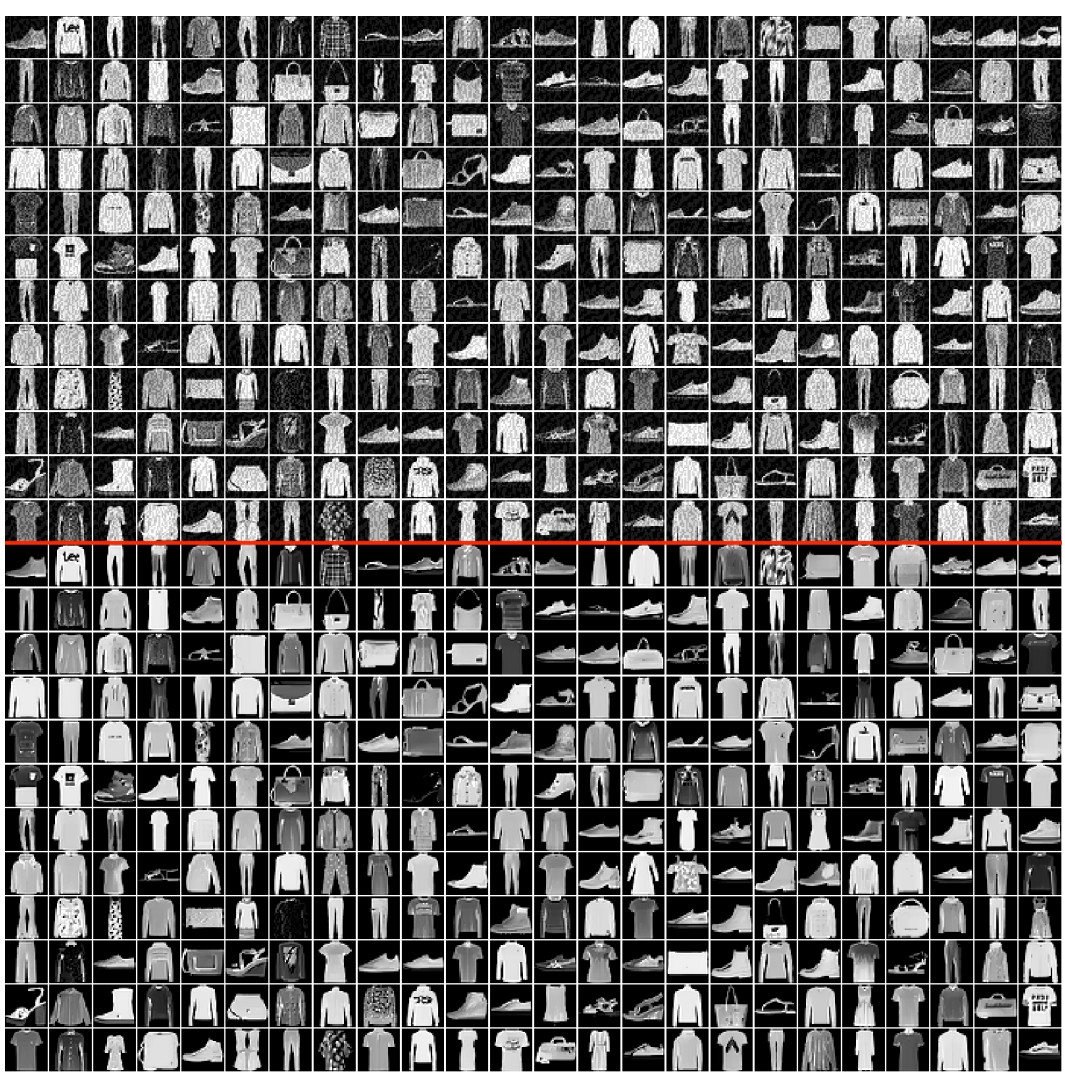

Figure 10: The upper part shows adversarial images generated from FGSM attack while the bottom part shows corresponding purified images after PixelDefend. Here $\epsilon_{\text{attack}} = 25$ and $\epsilon_{\text{defend}} = 32$.

## E.2 CIFAR-10

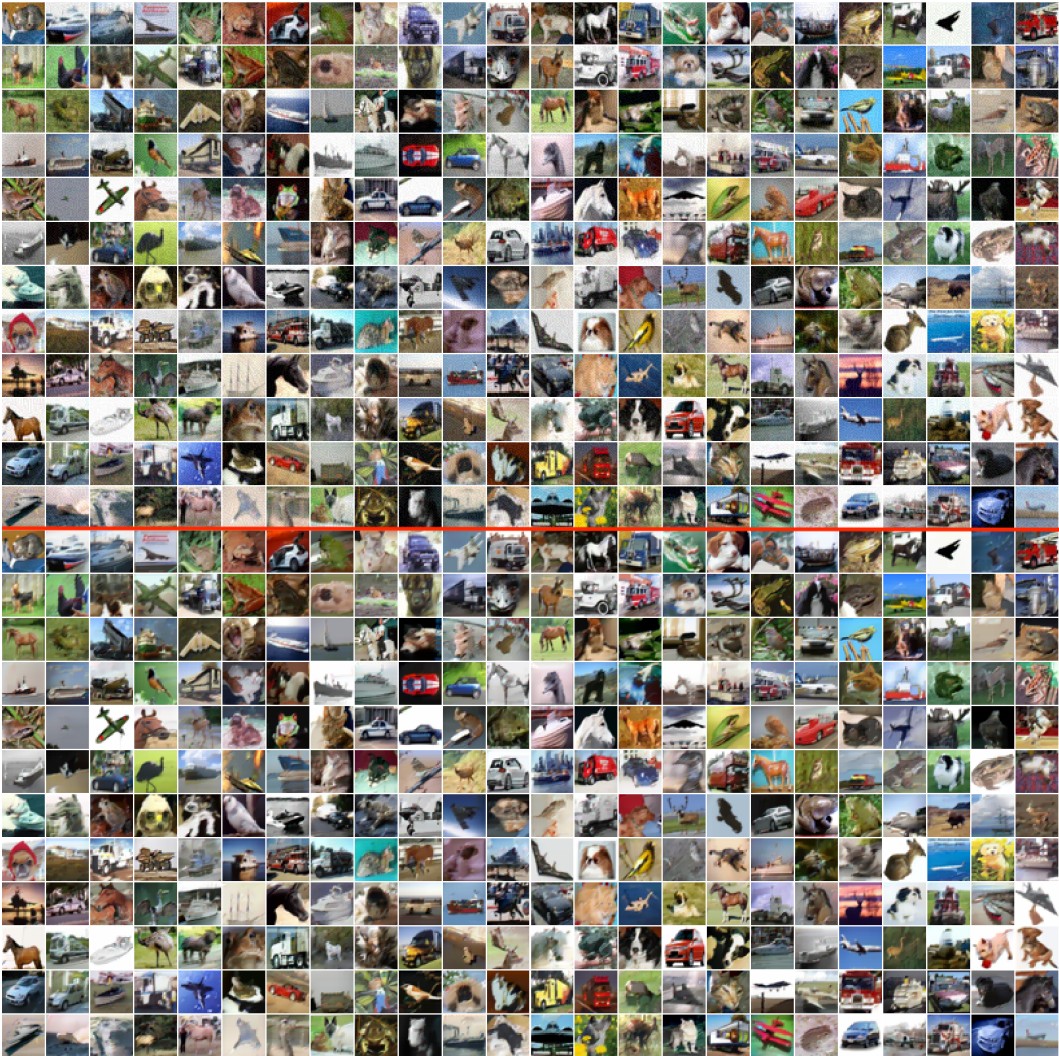

Figure 11: The upper part shows adversarial images generated from FGSM attack while the bottom part shows corresponding purified images by PixelDefend. Here $\epsilon_{\text{attack}} = 8$ and $\epsilon_{\text{defend}} = 16$.

