# OpenReview forum: "PixelDefend: Leveraging Generative Models to Understand and Defend against Adversarial Examples"
_ICLR.cc/2018/Conference — Accept (Poster)_

### Official Review · AnonReviewer1 · 2017-11-17
**Review of PixelDefend**

**Rating:** 7
**Confidence:** 4

**Review:**


I read the rebuttal and thank the authors for the thoughtful responses and revisions. The updated Figure 2 and Section 4.4. addresses my primary concerns. Upwardly revising my review.

====================

The authors describe a method for detecting adversarial examples by measuring the likelihood in terms of a generative model of an image. Furthermore, the authors prescribe a method for cleaning or 'santizing' an adversarial image through employing a generative model. The authors demonstrate some success in restoring images that have been adversarially perturbed with this technique.

The idea of using a generative model (PixelCNN) to assess whether a given image has been adversarially perturbed is a very interesting and understandable finding that may contribute quite nicely to the adversarial literature. One limitation of this method, however, is our ability to build successful generative models for high resolution images. However, I would be curious to know if the authors tried their method on high resolution images, regardless?

Major comments:
1) Cross validation. Figure 2a is quite interesting and compelling. It is not clear from the figure if the 'clean' (nor the other data for that matter) is from the *training* or  *testing* data for the PixelCNN model. I would *hope* that this is from the *testing* data indicating that these are the likelihood on unseen images?

That said, it would be interesting to see the *training* data on this plot as well to see if there are any systematic shifts that might make the distribution of adversarial examples less discernible.

2) Adversary to PixelCNN. It is not clear why a PixelCNN may not be adversarially attacked, nor if such a model would be able to guard against an adversarial attack. I am not sure how well viable of strategy this may be but it is worth understanding or addressing to determine how viable this method for guarding actually is.

3) Restorative effects of PixelDefend. I would like to see individual examples of (a) adversarial perturbation for a given image and (b) PixelDefend perturbation for that adversarial image. In particular, I would like to see how close (a) is the negative of (b). This would give me more confidence that this techniques is successfully guarding against the original attack.

I am willing to adjust my rating upward if the authors are able to address some of the points above in a substantive manner.

---

> ### Author Response · Authors · 2017-12-24
> **Response to AnonReviewer1**
>
> Thank you for your review! We would like to address each of your concerns point-by-point:
>
> Q: Generative models of high resolution images.
> A: We agree that testing PixelDefend on high resolution images is an important direction for future work. Although we haven’t tried PixelDefend on higher resolution images than CIFAR-10, we are hopeful that the PixelCNN can capture the approximate distributional properties to distinguish adversarial image even at resolutions where generating convincing samples becomes difficult. Experiments in the paper already provide one such piece of evidence: The samples given by PixelCNN on CIFAR-10 are already bad as judged by humans (see Figure. 9), while the samples for Fashion-MNIST (see Figure. 8) are almost indistinguishable from the training dataset. However, PixelDefend on CIFAR-10 is just as effective as PixelDefend on Fashion-MNIST.
>
> Q: Training or testing on Figure. 2a.
> A: These are indeed likelihoods of *testing* data on unseen images. We have revised the figure to add likelihoods of *training* data as well.
>
> Q: Adversary to PixelDefend
> A: We agree that the above arguments are not definitive and consider theoretical justifications to be an important direction for future research. In Section 4.4, we gave a discussion and provided some empirical results for an attack on PixelDefend. The arguments can be briefly summarized as:
>  * A naive attack of the PixelDefend purification process requires back-propagating thousands of repeated PixelCNN computations. This can lead to gradient vanishing problems, as validated by our experiments.
>  * Maximizing the PixelCNN density with gradient-based methods is very difficult (as shown in Figure. 5). Therefore such methods are not very amenable to generating adversarial images to fool a PixelCNN via gradient-based techniques.
>  * The PixelCNN is trained independent of labels. Therefore, the perturbation direction that leads to higher probability images has a smaller correlation with the perturbation direction that results in misclassification. This arguably makes attacking more difficult.
>
> Q: Restorative effects of PixelDefend.
> A: The goal of PixelDefend is not to undo the adversarial perturbations, but simply to avoid the problems they cause on the underlying classifier by pushing the image towards the nearest high probability mode of the distribution. These changes may not in general undo the adversarial changes, but (as our results show) will push the images towards the classification region for the original underlying class.

---

### Official Review · AnonReviewer2 · 2017-11-26
**A convincing new way to defend image models against adversarial examples.**

**Rating:** 7
**Confidence:** 4

**Review:**

The paper describes the creative application of a density estimation model to clean up adversarial examples before applying and image model (for classification, in this setup). The basic idea is that the image is first moved back to the probable region of images before applying the classifier. For images, the successful PiexlCNN model is used as a density estimator and is applied to clean up the image before the classification is attempted.

The proposed method is very intuitive, but might be expensive if a naive implementation of PixelCNN is used for the cleaning. The approach is novel. It is useful that the density estimator model does not have to rely on the labels. Also, it might even be trained on a different dataset potentially.

The con is that the proposed methodology still does not solve the problem of adversarial examples completely.

Minor nitpick: In section 2.1, it is suggested that DeepFool was the first optimization based attack to minimize the perturbation wrt the original image. In fact the much earler (2013) "Intriguing Propoerties ... " paper relied on the same formulation (minimizing perturbation under several constraints: changed detection and pixel intensities are being in the given range).

---

> ### Author Response · Authors · 2017-12-24
> **Response to AnonReviewer2**
>
> Thanks for pointing out our mistake of quoting DeepFool as the first optimization based attack to minimize the perturbation w.r.t. the original image. We have corrected it in the revised version.

---

### Official Review · AnonReviewer3 · 2017-12-02
**interesting experimental results, but no definitive argument**

**Rating:** 7
**Confidence:** 4

**Review:**

The authors propose to use a generative model of images to detect and defend against adverarial examples. White-box attacks against standard models for image recognition (Resnet and VGG) are considered, and a generative model (a PixelCNN) is trained on the same data as the classifiers. The authors first show that adversarial examples created by the white-box attacks correspond to low likelihood region (according to the pixelCNN), which first gives a classification rule for detecting adversarial examples.

Then, to turn the genrative model into a defensive algorithm, the authors propose to preprocess test images by approximately maximizing the likelihood under similar constraints as the attacker of images, to "project" adversarial examples back to high-density regions (as estimated by the generative model). As a heuristic method, the authors propose to greedily maximize the likelihood of the incoming images pixel-by-pixel, which is possible because of the specific form of the PixelCNN likelihood in the context of l-infty attacks. An "adaptive" version of the algorithm, in which the preprocessing is used only when the likelihood of an example is below a certain threshold, is also proposed.

Experiments are carried out on Fashion MNIST and CIFAR-10. At a high level, the message is that projecting the image into a high density region is sufficient to correct for a significant portions of the mistakes made on adversarial examples. The main result is that this approach based on generative models seems to work even on against the strongest attacks.

Overall, the idea proposed in the paper, using a generative model to detect and filter out spurious patterns that can appear in adversarial examples, is rather intuitive. The experimental result that adversarial examples can somehow be corrected by a generative model is also interesting. The design choice of PixelCNN, which allows for a greedy optimization seems reasonable in that setting.

Whereas the paper is an interesting step forward, the paper still doesn't provide definitive arguments in favor of using such approaches in practice. There is a significant loss in accuracy on clean examples (2% on CIFAR-10 for a resnet), and more generally against weaker opponents such as the fast gradient sign. Thus, in reality, the experiments show that the pipeline generative model + classifier is robust against the strongest white box methods for this classifier, but on the other hand these methods do not transfer well to new models. This somewhat weakens the result, since robustness against these methods that do not transfer well is achieved by changing the model.

---

> ### Author Response · Authors · 2017-12-24
> **Response to AnonReviewer3**
>
> Thank you for your review! Here is our answer to the following concern:
>
> Q: In reality, the experiments show that the pipeline generative model + classifier is robust against the strongest white box methods for this classifier, but on the other hand these (stronger attacking) methods do not transfer well to new models. This somewhat weakens the result, since robustness against these methods that do not transfer well is achieved by changing the model.
> A: Our assumption is that in real world circumstance we seldom have the option to change our classification model in response to the adversarial attack being used. We tend to think that the underlying classification model is generally going to be fixed and reasonably hard to change (i.e., in deployed autonomous driving cars), where the adversary can easily test out the system to decide which attack to use against it. Therefore it is important to defend against the strongest available attack.

---

### Decision · Program_Chairs · 2018-01-29
**ICLR 2018 Conference Acceptance Decision**

**Decision:**

Accept (Poster)

**Comment:**

The paper studies the use of PixelCNN density models for the detection of adversarial images, which tend to lie in low-probability parts of image space. The work is novel, relevant to the ICLR community, and appears to be technically sound.

A downside of the paper is its limited empirical evaluation: there evidence suggesting that defenses against adversarial examples that work well on MNIST/CIFAR do not necessarily transfer well to much higher-dimensional datasets, for instance, ImageNet. The paper could, therefore, would benefit from empirical evaluations of the defense on a dataset like ImageNet.